# HMOX1 Promotes Ferroptosis in Mammary Epithelial Cells via FTH1 and Is Involved in the Development of Clinical Mastitis in Dairy Cows

**DOI:** 10.3390/antiox11112221

**Published:** 2022-11-11

**Authors:** Quanwei Zhang, Xu Bai, Ting Lin, Xueying Wang, Bohao Zhang, Lijun Dai, Jun Shi, Yong Zhang, Xingxu Zhao

**Affiliations:** 1College of Life Science and Technology, Gansu Agriculture University, Lanzhou 730070, China; 2College of Veterinary Medicine, Gansu Agriculture University, Lanzhou 730070, China; 3Gansu Key Laboratory of Animal Reproductive Physiology and Reproductive Regulation, Lanzhou 730070, China

**Keywords:** DIA proteomics, HMOX1, FTH1, MAC-T, ferroptosis, mastitis

## Abstract

Ferroptosis is associated with inflammatory diseases as a lethal iron-dependent lipid peroxidation; its role in the development of clinical mastitis (CM) in dairy cows is not well understood. The aim of this study was to identify differentially expressed proteins (DEPs) associated with iron homeostasis and apoptosis, and to investigate further their roles in dairy cows with CM. The results suggested that ferroptosis occurs in the mammary glands of Holstein cows with CM. Using data-independent acquisition proteomics, 302 DEPs included in 11 GO terms related to iron homeostasis and apoptosis were identified. In particular, heme oxygenase-1 (HMOX1) was identified and involved in nine pathways. In addition, ferritin heavy chain 1 (FTH1) was identified and involved in the ferroptosis pathway. HMOX1 and FTH1 were located primarily in mammary epithelial cells (MECs), and displayed significantly up-regulated expression patterns compared to the control group (healthy cows). The expression levels of HMOX1 and FTH1 were up-regulated in a dose-dependent manner in LPS induced MAC-T cells with increased iron accumulation. The expression levels of HMOX1 and FTH1 and iron accumulation levels in the MAC-T cells were significantly up-regulated by using LPS, but were lower than the levels seen with Erastin (ERA). Finally, we deduced the mechanism of ferroptosis in the MECs of Holstein cows with CM. These results provide new insights for the prevention and treatment of ferroptosis-mediated clinical mastitis in dairy animals.

## 1. Introduction

Ferroptosis is a cell death form characterized by lethal iron-dependent lipid peroxidation [1]. The typical morphology in small mitochondria includes condensed mitochondrial membrane densities, reduction or disappearance of mitochondria crista, and outer mitochondrial membrane rupture [2]. Biochemically, ferroptosis is characterized by chemical or mutational inhibition of the cystine/glutamate antiporter, culminating in the accumulation of reactive oxygen species (ROS) in the form of lipid hydroperoxides [3]. As in other apoptotic or non-apoptotic cell death, ferroptosis is regulated by functional genes or proteins such as glutathione peroxidase 4 (GPX4), solute carrier family 7 member 11 (SLC7A11) and nuclear factor erythroid 2-like (NRF2) [2]. The regulatory mechanisms of these genes in ferroptosis can be classified into biological and chemical categories, that activate two major pathways, the extrinsic/transporter and the intrinsic/enzymatic pathways [2,4,5]. For instance, the dysregulation of either extrinsic or intrinsic iron metabolism, especially iron overload, is an important cause of ferroptosis. The excessive peroxidation of phospholipid (PL) membranes rich in polyunsaturated fatty acids (PUFAs), especially in arachidonic acid (AA) metabolism, is caused by ferroptosis via an iron-dependent (by Fenton reactions) and lipoxygenases (LOXs) enzymatic mechanisms [6,7,8]. Blocking transportation of cystine-glutamate antiporter system Xc- causes excessive intracellular ROS and lethal lipid peroxidation, which caused ferroptosis via the GPX4-dependent pathway [9,10]. Ferroptosis can be activated due to excessive iron metabolism, abnormal amino acid and lipid metabolism and participates in normal development, homeostasis, and the prevention and mitigation of diseases in mammals. Emerging evidence suggests that iron-dependent oxidative stress and lipid peroxidation are also common features of ferroptosis and inflammatory diseases [1,11]. Thus, understanding the molecular mechanisms of ferroptosis in inflammatory diseases will help to investigate pathogenic mechanisms and provide new diagnostic and therapeutic approaches or drugs.

Clinical mastitis (CM), one of the most important and costly diseases in the dairy industry, is an inflammatory disease of mammary glands in dairy animals. Increasing research has been focused on the prevention and treatment of CM from different perspectives, such as developing new drugs, treatment approaches and vaccines [12,13,14]. However, CM remains a problem that plagues the global dairy industry. Due to the high prevalence of mastitis and the subsequent overuse of antibiotics, novel compounds or new classes of antibiotics are desired. Increasingly, evidence suggests that the destruction of internal environmental homeostasis caused by pathogen infection and immune system disorders, as well as a lack of mineral elements are the crucial reasons for CM in dairy animals [15,16,17,18]. Coincidentally, the competition for iron in the battle between microbials and host cells underlies infection and inflammation development [1]. Ferroptosis cells can be recognized by immune cells that trigger a range of inflammatory or specific responses [4]. In addition, it is known that, in mastitis, inflammation is accompanied by phospholipase activation that produces AA from membrane phospholipids [19]. AA metabolites are necessary for cyclooxygenase and LOXs enzymatic systems, and the metabolism of AA through three metabolic pathways into precursors of bioactive proinflammatory mediators is the core inflammatory pathway [4,5,19]. These characteristics of mastitis are highly consistent with those of ferroptosis. Due to a lack of relevant studies, whether these factors are the main determinants behind ferroptosis in the mammary glands in cows, and the link between these factors and ferroptosis, is poorly understood. Understanding the target functional proteins, molecular mechanisms and signaling pathways of ferroptosis in the mammary glands may provide new diagnostic and therapeutic approaches to regulateing cell survival and death in dairy animals.

The goals of the current study were to identify candidate differentially expressed proteins (DEPs) or target molecules associated with ferroptosis and iron homeostasis in cows with CM using bioinformatics analysis based on the data-independent acquisition (DIA) proteomics and to further demonstrate the relevant mechanisms in the mammary epithelial cells (MECs). The results will provide a better understanding of the role of ferroptosis in CM.

## 2. Materials and Methods

### 2.1. Sample Preparation and Collection

Holstein cows (6 years of age) from a commercial farm (Wuzhong City, Ningxia Hui Autonomous Region, China) were evaluated for clinical mastitis. The udder examination was performed according to the criteria for veterinary clinical diagnosis, including redness, temperature, swelling and pain [20]. Milk samples (20–25 mL) from the Holstein cows were collected for somatic cell count (SCC) as described previously [21]. After screening, Holstein cows with clinical symptoms and SCC ≥ 13 × 10^5^ cells/mL were selected as the clinical mastitis (CM) group (*n* = 3) and the healthy Holstein cows without clinical symptoms and SCC ≤ 1 × 10^5^ cells/mL were selected as the control (Con/C) group (*n* = 3). For the validation experiments, three cows were selected for each of the C and CM groups. The Holstein cows were transferred to a slaughterhouse, where the mammary gland tissues were collected.

The mammary gland tissues were fixed using 4% paraformaldehyde or 2.5% glutaraldehyde for morphological observation and location analysis using transmission electron microscopy (TEM), Perls plus DAB staining, ROS detection, hematoxylin–eosin (H&E) staining, immunohistochemistry (IHC) and immunofluorescence (IF) staining. Some of the tissues were immediately stored at −80 °C for proteomics, mRNA and protein expression pattern analyses using data-independent acquisition (DIA), quantitative real time polymerase chain reaction (qRT-PCR) and Western blot. All samples were collected in accordance with the ethical guidelines approved by the Animal Care Commission of Gansu Agricultural University (GSAU-Eth-LST-2021-003).

### 2.2. Cell Culture and Treatment

Immortalized mammary epithelial cells (MAC-T) were purchased from the ATCC Corporation (Beijing, China). The MAC-T cells were maintained in different dishes in a humidified atmosphere of 5% CO_2_ at 37 °C in DMEM culture medium (Gibco, Grand Island, NY, USA) containing 10% fetal bovine serum (FBS, Gibco). The cells were grown to 90% confluency prior to passage or treatment. 12 h before treatment, the culture medium was shifted to DMEM without FBS and phenol (Gibco) for 12 h. The MAC-T cells were treated with or without DMSO (0.1%), LPS (0–200 μg/mL, Solarbio, Beijing, China) and ferroptosis agonist Erastin (ERA, 0.1–10 μM, Selleck, Houston, TX, USA) for 24 h. Each treatment was replicated at least in triplicate.

### 2.3. Morphological Observation and Immunological Staining

TEM was used in order to observe the morphologic changes of the mitochondria in the mammary glands of the C and CM groups, and the treated MAC-T cells as described previously [22,23]. Briefly, after being fixed, rinsed, dehydrated, the tissues were embedded in paraffin or agarose for ultrathin sections (60–80 nm in thickness) and then were stained with uranyl acetate and lead citrate. Images were captured using a Hitachi HT7700 TEM (Hitachi Science and Technology, Minto-Ku, Japan). Perls plus DAB, ROS, H&E, IHC and IF staining assays were carried out as described previously [24,25,26,27,28]. Briefly, fixed tissues were embedded in paraffin (Solarbio, Beijing, China) and cut into 5 μm thick sections using a microtome (Leica, Shanghai, China). The sections were deparaffinized in xylene and rehydrated. Staining was carried out as described previously [24]. The antibodies and dilution ratio used in the present study are presented in Appendix A. The images were captured using an Olympus microscope BX43 (Olympus, Tokyo, Japan). Integral optical density (IOD) values of the images were quantified and scanned using Image-Pro Plus 6.0 (Media Cybernetics Co., Rockville, MD, USA). All staining assays and data collection were performed at least in triplicate.

### 2.4. DIA Proteomics and Bioinformatics

Total protein was extracted from the mammary gland tissues of the C and CM groups. Protein extraction and digestion, liquid chromatography separation, mass spectrometry, and proteomics data analysis were carried out as described previously [29,30] with slight modifications. Raw DIA data were processed and analyzed via Spectronaut X (Biognosys AG, Switzerland) with default settings to generate an initial target list. Spectronaut X was set up to search the Bos taurus database (NCBI_GCF_002263795.1). *Q*-value (FDR) cutoff on precursor and protein level was set to 1%. All precursors passing these filters were used for quantification. The average top three filtered peptides which passed the 1% *Q*-value cutoff were used to calculate the major group quantities. After applying the Student’s *t*-test, proteins with *Q* < 0.05 and Absolute log_2_ ratio > 0.58 were identified as differentially expressed proteins (DEPs). The DIA proteomics sequence was deposited in ProteomeXchange with accession number IPX0003382000/PXD028100. The DEPs (detailed in Appendix A) generated from the raw DIA proteomics sequence data were used for Gene Ontology (GO) and Kyoto Encyclopedia of Genes and Genomes (KEGG) pathway annotation using R packages as described previously [31]. The GO terms and KEGG pathways with *Q* < 0.05 were considered to be statistically significant. In the present study, we focused on the DEPs and the biological processes (*p* < 0.05 and *Q* < 0.05) with GO terms associated with apoptosis and ion homeostasis. After overlapping and intersecting the DEPs, pathways including HMOX1 and the DEPs (*p* < 0.05 and *Q* < 0.05) were also selected for further study. Heat maps, circos plots, bubble diagrams, volcano plots, and Venn and Sankey diagrams regarding the functional analysis of the DEPs were drawn using R packages and the OmicShare tools online platform [24,31]. Protein-protein interaction (PPI) networks of the candidate DEPs, GO terms and pathways were constructed using STRING v 10.0, Cytoscape 2.8.1, including ClueGO, and Ingenuity Pathway Analysis (IPA) as described previously [32,33].

### 2.5. ROS and Cell Viability Assays

MAC-T cells were treated with or without DMSO (0.1%), LPS (0–200 μg/mL) and Erastin (0.1–10 μM) for 24 h. The ROS was measured using a ROS Assay Kit (ROS, Servicebio, Wuhan, China) in these four groups according to the manufacturer’s instructions. The relative fluorescence intensity of ROS in the MAC-T cells of the four groups was quantified and scanned using Image-Pro Plus 6.0 (Media Cybernetics Co., Rockville, MD, USA). Group DMSO was used as the control. At least three visual fields were randomly selected from each slice, and all optical density scans were repeated at least in triplicate. The viability of MAC-T cells was measured using a Cell Counting Kit 8 (CCK8, Bimake, Shanghai, China) according to the manufacturer’s instructions. Briefly, the cells were seeded in 96-well plates at 1 × 10^4^ cells per well and incubated for 24 h. The cells were treated with different concentrations of LPS and Erastin for 24 h. 10 µL of CCK8 was added to the cells and allowed to incubate for 2 h before measuring the OD at 450 nm with a ReadMax 1900 microplate reader (Shanghai Flash, Shanghai, China). The cell viability experiments were performed at least in triplicate.

### 2.6. RNA Isolation, cDNA Synthesis and qRT-PCR

Total RNA was extracted from the mammary gland tissues of the C and CM groups, and the MAC-T cells using a FastPure RNA isolation kit (Vazyme, Nanjing, China) according to the manufacturer’s instructions. RNA was quantified on a NanoDrop-8000 (Thermo Fisher Scientific, Waltham, MA, USA) and RNA integrity was assessed via denaturing formaldehyde agarose gel (1%) electrophoresis (Biowest Regular Agarose, Castropol, Spain). 1 μg of total RNA was subjected to reverse transcription to complementary single-stranded DNA (cDNA) using Thermo RT Kit (BioTeke, Beijing, China). cDNA was synthesized via reverse transcription PCR and was performed according to the manufacturer’s instructions. The relative expression levels of Heme oxygenase-1 (HMOX1) and ferritin heavy chain 1 (FTH1) mRNA in the mammary gland tissues and MAC-T cells were detected using qRT-PCR. The expression of *GAPDH* was used as an endogenous control. qRT-PCR primers (Appendix A) were designed using premier 5.0 software [24] and synthesized by Qinke Biotech Co., Ltd. (Shanxi, China). qRT-PCR was performed on a LightCycler 96 real-time PCR system (Roche, Switzerland). The procedures and result calculations were carried out as described previously [24,25,28].

### 2.7. Western Blot

The relative expression levels of HMOX1 and FTH1 protein in the mammary gland tissues of C and CM groups and the MAC-T cells were measured using Western blot. Total protein was extracted from tissue and cell samples using RIPA cell lysis reagent (Solarbio, Beijing, China). The procedures were carried out as described previously [28]. The primary antibodies were incubated in different dilution factors at 4 °C overnight (Appendix A). Integrated optical density (IOD) values of the bands were scanned and quantified using Image-Pro Plus 6.0 software. The expression level of β-actin was used as an endogenous control. All immuno-blot assays were performed at least in triplicate.

### 2.8. Co-Immunoprecipitation

The protein interaction between HMOX1 and FTH1 was analyzed using Co-immunoprecipitation (Co-IP) assays following the manufacturer’s instructions. The immunoprecipitation experiments were carried out as described previously [34]. Briefly, the MAC-T cells were treated with or without LPS (100 μg/mL) for 24 h. Total protein from the cells was extracted and purified using a Co-IP commercial kit (Thermo Scientific Pierce Inc., Beijing, China). PierceTM Protein A/G Agarose beads and corresponding antibodies (anti-HMOX1, anti-FTH1, and anti-IgG) were used for immunoprecipitation. The resulting immuno-complex was analyzed using Western blot analysis. All immuno-blot assays were performed at least in triplicate.

### 2.9. Statistical Analysis

No assumptions for normality of the DIA proteomics data and validated data of the tissues were made due to the small sample size. These variables such as Fe^3+^, ROS, qRT-PCR, and Western blot data in the C and CM groups were expressed as the median (range), and analyzed using SPSS 22.0 software (SPSS Inc., Chicago, IL, USA) with the Wilcoxon rank sum procedure (α = 0.05). The data generated from the cellular samples were presented as the mean ± SEM, and statistically analyzed using SPSS version 22.0. qRT-PCR and Western blot data between two groups were analyzed using the Student’s *t*-test and data between multiple groups was analyzed using one-way ANOVA. Graphs were constructed using GraphPad Prism 9.0 (GraphPad Software Inc., San Diego, CA, USA). Statistical significance was set to *p* < 0.05.

## 3. Results

### 3.1. Observation of the Phenotypes Related to Ferroptosis in the Mammary Glands of Holstein Cows

Phenotypes related to ferroptosis, including variations in the mitochondria, ferric ion and ROS, were observed in the mammary gland tissues of the C and CM groups (Figure 1). TEM results showed that the crista and membranes of the mitochondria were intact and clear in healthy mammary glands of the C group (Figure 1A1), while vanishing mitochondria crista and condensing of the mitochondrial membrane densities were observed in the CM group (Figure 1A2). H&E results displayed that intact alveoli with neatly arranged MECs in the C group (Figure 1B1). However, the alveoli were collapsed, and coupled with plentiful exfoliated MECs and infiltrated neutrophils in the mammary glands of the CM group (Figure 1B2). Perls plus DAB staining revealed that ferric iron (Fe^3+^) was dyed rarely in the C group (Figure 1C1), whereas substantial ferric iron appeared as brown masses in the pathological alveoli of the CM group (Figure 1C2). Compared to the control group, the relative levels of ferric iron were more than 68.92-fold changed in the CM group (Figure 1D). ROS fluorescence staining suggested that the positive ROS signals in the CM group were obviously stronger than those in the C group (Figure 1E,F). Statistical analysis showed that the relative staining levels of ROS in the CM group were 2.12-folds higher than those in the C group (Figure 1G). These results revealed that ferroptosis was present in the mammary glands of the Holstein cows with CM, particularly in the alveoli. Our results suggested that ferroptosis in the mammary glands has a close relationship with the development of CM in dairy cows.

### 3.2. Identification of Candidate DEPs Associated with Ion Homeostasis, Cell Proliferation and Apoptosis Based on the GO Terms of DIA Proteomics

A total of 76,255 precursors, 63,103 peptides, 7618 protein groups and 20,355 proteins were identified with a filter criteria of FDR < 0.01, respectively. Compared to the C group, a total of 3739 DEPs and 819 significant biological processes were identified from the DIA proteomics analysis (Appendix A). A total of 68 pathways with *Q* < 0.05 were also identified according to these DEPs. In the current study, we focused on the biological processes and candidate DEPs associated with ion homeostasis, cell proliferation and apoptosis based on biological process of ferroptosis (Figure 2). A total of 45 DEPs from three biological processes related to ion homeostasis and 288 DEPs from eight biological processes related to cell proliferation and apoptosis were selected as candidate DEPs (Figure 2A). The Venn diagram showed that HMOX1 was shared in the 11 biological processes (Figure 2B), which suggested HMOX1 plays crucial roles in these biological processes. After overlapping the repeat DEPs in different biological processes, a total of 302 DEPs, including 235 up-regulated and 66 down-regulated DEPs, were identified as candidate DEPs associated with ion homeostasis and cell development (Appendix A and Figure 2C). The interaction network of the biological processes was constructed using these 302 DEPs, and the results indicated that there were no direct interactions among ion homeostasis, cell apoptosis and proliferation. However, HMOX1 was the only DEP connected with ion homeostasis, cell proliferation and apoptosis (Figure 2D). Subsequently, using HMOX1 as the core protein, the PPI network was constructed again using the 302 DEPs, and the results suggested that 100 DEPs, including 89 up-regulated and 11 down-regulated DEPs, interacted directly with HMOX1 protein (Figure 2E). These results indicated that HMOX1 was a crucial candidate DEP associated with ion homeostasis, cell apoptosis, and ferroptosis.

### 3.3. Identification of the Pathways and Candidate DEPs Associated with HMOX1 Based on the KEGG Pathway Analysis of DIA Proteomics

The pathways and candidate DEPs involving HMOX1 were selected for identifying the potential mechanisms of HMOX1 activity in the mammary glands of the Holstein cows with and without CM (Figure 3). A total of 320 DEPs, selected from the DIA proteomics analysis and included in nine pathways, were identified (Appendix A and Figure 3A). Some of the pathways, such as metabolic pathways, mineral absorption and ferroptosis pathways, have a close relationship with homeostasis, in particular with ion metabolic balance. The IPA results showed that HMXO1 participated directly in the nine pathways, particularly the ferroptosis and mineral absorption pathways, and also interacted with cancer and metabolic pathways (Figure 3B). In the present study, we focused on the ferroptosis pathway. The heat map of the ferroptosis pathway showed that a total of 11 DEPs were differently expressed in the mammary glands of the C and CM groups, specifically the FTH1 and HMOX1 proteins with a significant difference (*p* < 0.01, Figure 3C). The Sankey diagram was created to reflect the relationship among the shared DEPs (Appendix A), the nine pathways and the 11 biological processes associated with ion homeostasis, cell proliferation and apoptosis. The results revealed that the pathways and the biological processes were enriched with many DEPs, including ten DEPs such as BAD, BAX, STAB3, FTH1 and HMOX1 (Figure 3D). Our results showed that HMOX1, as a crucial DEP, was involved in ion homeostasis, cell proliferation and apoptosis in the mammary glands of the Holstein cows with CM.

### 3.4. The Distribution and Expression Pattern Analysis of HMOX1 and FTH1 in Mammary Glands of the Holstein Cows

The immune-positive reaction of HMOX1 and FTH1 protein and the expression pattern of HMOX1 and FTH1 mRNA and protein were measured in the mammary glands of the C and CM groups using different assays (Figure 4). IHC staining results showed that the positive staining of HMOX1 and FTH1 proteins were distributed mainly in the MECs of the C and CM groups (Figure 4A,B). The stronger positive staining reaction of HMOX1 and FTH1 proteins were distributed in the inflammatory reaction area of the CM group and were higher than those of the C group. The control group had no positive staining reaction of HMXO1 and FTH1 proteins (Figure 4C). For IF staining, CK18, a specific marker of MECs, was obviously present in the mammary alveoli (MA), especially in the intact alveoli of the C group (Figure 4E). Compared to the C group, the positive IF signals of HMOX1 and FTH1 proteins were in the MECs, and displayed a stronger expression (Figure 4F,G). After merging these signals, we found that CK18, HMOX1 and FTH1 proteins were co-located in the MECs of the C and CM groups (Figure 4H). In addition, it was observed that the neutrophils were present in large numbers in the MA of the CM group. HMOX1 and FTH1 also presented with stronger expression levels in the MA of the CM group. The qRT-PCR results showed that the relative expression levels of HMOX1 and FTH1 mRNA in the CM group were significantly up-regulated compared to the C group (Figure 4I,J). Western blot results displayed that the HMOX1 and FTH1 proteins were indicated in the C and CM groups. The bands of HMOX1 and FTH1 proteins were stronger in the CM group than in the C group (Figure 4K). The Western blot IOD values suggested that the relative expression levels of HMOX1 and FTH1 proteins in the CM group were significantly higher compared to the C group (Figure 4L,M). These results revealed that up-regulation of HMOX1 and FTH1 mRNA and proteins in the MECs is the potential mechanism that positively regulates the development of CM in dairy cows.

### 3.5. Up-Regulation of HMOX1 and FTH1 via LPS Induced Inflammation in MAC-T Cells

In order to verify the link between HMOX1 and FTH1, an inflammation model was induced using different concentrations of LPS in MAC-T cells (Figure 5). Co-localization results displayed that the HMOX1, FTH1 and CK18 proteins were co-located in the cytoplasm of MAC-T cells (Figure 5A). CCK8 results showed that the viability of the MAC-T cells was significantly decreased after treatment with increasing concentrations of LPS (*p* < 0.05) (Figure 5B). The cell morphology results (Figure 5C) showed that MAC-T cells (Control, treated without LPS) formed a pebble-like monolayer aggregation and were arranged closely, which was consistent with the morphological characteristics of epithelial cells (Figure 5C1). However, LPS-treated (100 μg/mL) MAC-T cells were loosely arranged with fusiform or irregular shapes, displayed inhibited cell growth, and showed decreased vitality (Figure 5C2). Perls Prussian blue staining results indicated that the positive staining of Fe^3+^ (brown) was only slightly present in the C group (Figure 5D1), while it was substantially present in the LPS group (Figure 5D2). Compared to the C group, the relative expression levels of HMOX1 and FTH1 mRNA were significantly up-regulated in a dose-dependent manner with increasing LPS concentrations (0.01 to 200 μg/mL) (Figure 5E,F). The HMOX1 and FTH1 proteins were detected in MAC-T cells treated with different concentrations of LPS (Figure 5G). The IOD values revealed that HMOX1 and FTH1 proteins were also up-regulated with an increase of LPS concentrations, especially treatment with 1 to 200 μg/mL LPS (Figure 5H,I). These results suggested that the ferroptosis that could be induced in MAC-T cells using LPS depended on the up-regulation of HMOX1 and FTH1.

### 3.6. HMOX1 Promotes Ferroptosis via the FTH1 Dependent Pathway in LPS Induced Inflammation in MAC-T Cells

The positive staining of ferric iron (brown) was apparent in MAC-T cells treated with LPS (100 μg/mL) and the ERA (5 μM) groups, while it was not present in the C (DMSO, 0.1% *v/v*) and NC groups (Figure 6A). TEM results suggested that the cristae and membranes of the mitochondria were intact and clear in the MAC-T cells of the DMSO and NC groups, while vanishing mitochondrial cristae and condensing of the mitochondrial membrane densities were observed in the MAC-T cells of the LPS and ERA groups (Figure 6B). ROS results displayed that the positive fluorescence signals were observed in four groups. The ROS signals in the ERA and LPS groups were obviously stronger than that in DMSO and NC groups (Figure 6C). Compared to the DMSO and NC groups, the relative fluorescence intensity of ROS was up-regulated in the LPS and ERA groups with a significant difference (Figure 6D). The CCK8 results showed that the viability of the MAC-T cells was significantly decreased after treatment with differing concentrations of ERA (0.1 to 10 μM), compared to the viability of the NC group. There was no significant difference in the MAC-T cells treated with 5 or 10 μM ERA groups (Figure 6E). Compared to the C group, the relative expression levels of HMOX1 and FTH1 mRNA were significantly up-regulated in MAC-T cells treated with LPS or ERA agonist (Figure 6F,G). In addition, compared to the LPS group, the relative expression levels of HMOX1 and FTH1 mRNA were significantly up-regulated in MAC-T cells treated with ERA agonist (*p* < 0.05). The immuno-blot results showed that the HMOX1 and FTH1 proteins were present in four groups, but especially so in the LPS and ERA groups (Figure 6H). Compared to the C and NC groups, HMOX1 and FTH1 proteins were significantly (*p* < 0.01) up-regulated in the LPS and ERA groups (Figure 6I,J). Co-IP results revealed that immunoblotting (IB) signals of FTH1 or HMOX1 could be measured in MAC-T cells which were treated with and without LPS (Figure 6K), but no FTH1 and HMOX1 signals were noted in the protein pull-down by IgG antibody. These results suggested that FTH1 and HMOX1 proteins were directly interacting to regulate ferroptosis in the LPS induced inflammation model in MAC-T cells.

### 3.7. The Deduced Mechanism of HMOX1 and FTH1 in the MECs of Holstein Cows with CM

The important proteins associated with ferroptosis were selected from the 3739 DEPs identified by DIA proteomics analysis. A total of 13 proteins (Appendix A), including three DEPs related to amino acid metabolism, three related to iron metabolism, and seven DEPs related to lipid metabolism in ferroptosis, were selected (Figure 7A). Among these DEPs, FTH1, HSPB1, HMOX1, ACSF2 and CARS showed significantly different expression (*p* < 0.05 and *Q* < 0.05) compared to that in the C group. FTH1 and HMOX1 were notable and had a log_2_(FC) > 1.5. This suggested that ferroptosis in the mammary glands of Holstein cows with CM is dependent on the iron metabolism performed by HMOX1 and FTH1. The mechanism of ferroptosis in mammary glands of the Holstein cows with CM was deduced based on these results (Figure 7B). The source of excessive ferric iron in the mammary glands of the Holstein cows with CM was classified into two ways. First, absorbed ferric iron in the intestinal epithelial cells combined with transferrin receptor 1(TFR1) and transported into the MECs for maintaining the dynamic iron homeostasis. Second, ferric iron in senescent RBCs were degraded by HMOX1 and further phagocytized by macrophages, resulting in iron recycling or accumulation of ferric iron. Excessive accumulated ferric iron in the MECs mediates the Fenton reaction, which generates large amounts of hydroxyl radicals and initiates liposome peroxidation. This, in turn, results in ROS production and activation of ferroptosis in the mammary glands via the FTH1 dependent pathway.

## 4. Discussion

Ferroptosis is a non-apoptotic cell death pathway and plays an important role in many diseases, especially inflammatory diseases [1,4,5]. However, the phenomenon and regulatory mechanism of ferroptosis in the mammary glands of Holstein cows with CM has remained under-characterized. In the present study, we observed the phenotypes related to ferroptosis and the results confirmed that ferroptosis was present and caused by the accumulation of ferric iron in the mammary glands of Holstein cows with CM (Figure 1). The excessive accumulation of ferric iron is possibly caused by the competition of pathogens for iron and senescent RBC degradation. Iron, an essential nutrient for bacterial growth, is utilized by chelation mechanisms for bacteria [35]. The production of siderophores such as enterobactin is induced under low iron level conditions, aiding bacteria in their competition with the host for scarcely available iron [36,37]. Bacterial siderophores are positively associated with the ROS production of polymorphonuclear neutrophilic granulocytes [38]. This mechanism of iron competition in pathogens is the possible cause of CM by pathogens in dairy animals. On the other hand, RBCs escaping from blood vessels are phagocytized and degraded, resulting in heme in the extracellular milieu or the MA in mammary glands of the Holstein cows with CM. The iron released from heme participates in essential biological processes, even exerting several deleterious effects, such as damage to lipids and proteins, and inducing excessive ROS [39,40]. Along with its oxidative dependent effects, heme also activates innate immune responses and inflammation [40]. This may be one of the reasons that the accumulation of ferric iron in the mammary glands leads to the development of CM in dairy cows.

In order to illustrate the mechanism of ferric iron accumulation in mammary glands of the Holstein cows with CM, bioinformatics analysis was performed according to the 3739 DEPs identified from DIA proteomics analysis. Compared to other proteomics approaches, DIA proteomics is a more comprehensive, repeatable and precise approach that has been utilized in studying the mechanisms of various diseases [41,42]. Because accumulated evidence identifies ion channels as essential regulators of apoptosis [43], and considering the features of ferroptosis, we focused on the biological processes associated with ion homeostasis and apoptosis. High iron concentrations in plasma exceeding the iron-binding capacity of transferrin can damage cells and tissues through the generation of ROS and regulation of apoptosis [44]. In the present study, a total of 11 biological processes, including 302 DEPs, were selected based on the DIA proteomics data, and HMOX1 in particular (Figure 2). HMOX1, as a key cytoprotective gene and enzyme, is under complex regulation and can be up-regulated markedly by heme, iron and other factors [45]. Increasingly, studies have demonstrated that HMOX1 is associated with apoptosis, proliferation and inflammation [46]. Of course, some candidate DEPs, such as PTPRC, BAX, CAV1 and PML, are also associated with iron metabolism and apoptosis [47,48,49]. However, the functions and mechanisms of these DEPs are still incompletely known and require further study to elucidate. Subsequently, we focused on the pathways involving HMOX1. The results suggested that metabolic pathways, mineral absorption and ferroptosis pathways were interacting with HMOX1 (Figure 3). It was revealed that HMOX1 participated in the biological processes related to these pathways. For instance, up-regulation of HMOX1 contributes to endothelial cell ferroptosis by promoting iron overload, ROS generation and lipid peroxides [50]. It has also been demonstrated that up-regulated HMOX1 can induce ferroptosis by increasing ROS and intracellular ferric iron levels via suppression of GPX4 expression [51]. Furthermore, bioinformatics analysis suggested that HMOX1 and FTH1 were crucial DEPs in the ferroptosis pathway. It has been demonstrated that FTH1 is an important protein in iron dependent ferroptosis [5,6]. Hence, we hypothesized that ferroptosis in the mammary glands of Holstein cows with CM was dependent on concentration variations of FTH1 and HMOX1 proteins.

IHC and IF results confirmed that the FTH1 and HMOX1 proteins were co-located mainly in the MECs (Figure 4), indicating that they may be associated with MEC functions. MECs have important effects on innate immunity, inflammation and defense against mastitis [52]. The results of expression pattern analysis hinted that up-regulation of HMOX1 and FTH1 were positively related with the occurrence and development of CM and ferroptosis in dairy cows. Compelling evidence indicates that ferroptosis plays an important role in inflammatory diseases [11]. LPS, as a Gram-negative bacterial endotoxin, is frequently used for constructing various animals and cellular inflammation models. We established an inflammatory model in MAC-T cells using LPS (Figure 5). The expression levels of HMOX1 and FTH1 mRNA and proteins were up-regulated, compared to the C group, and were dose-dependent with the levels of iron accumulation. The results indicated that ferroptosis was activated in LPS-induced MAC-T cells. It was also confirmed that ferroptosis was activated in LPS-induced myofibroblasts [53]. FTH1, as a subunit of ferritin, can accommodate 4500 ferrous iron atoms and oxidize ferrous iron to ferric iron in an oxygen-dependent manner [54]. The results indicated that ferroptosis was activated in LPS-induced MAC-T cells by HMOX1 and FTH1 in an iron metabolism dependent manner. ERA, as an agonist of ferroptosis, was used as a positive control for verification of the mechanism in LPS-induced MAC-T cells. Similar to the results seen in ERA treated cells, the typical characteristics of ferroptosis including abnormal mitochondria, accumulated ferric iron and excessive ROS were present in the LPS- and ERA-treated MAC-T cells (Figure 6). The expression levels of HMOX1 and FTH1 mRNA and proteins in LPS-treated MAC-T cells were up-regulated, but at significantly lower levels than the ERA group. This may be due to the specificity of ERA and LPS in ferroptosis. These results further conformed that ferroptosis can be induced by LPS or ERA in the MECs. Co-IP results confirmed that FTH1 and HMOX1 proteins interacted directly to regulate ferroptosis. Previous studies have demonstrated that HMOX1 accelerates Erastin-induced ferroptosis through enzymatic degradation of the heme [55,56]. In addition, HMOX1 has been shown to regulate cellular iron homeostasis of both free and chelated iron, which contributes to the accumulation of ferrous ions [57]. Taking the results together, ferroptosis was deduced in the MECs of Holstein cows with CM (Figure 7). Ferroptosis relied on the degree of iron metabolism that is regulated by HMOX1 and FTH1, but not GPX4 or SLC7A11 or other ferroptosis activators. Previous studies also demonstrated that different ferroptosis mechanisms occur in various cells and tissues [2,4,5,53]. These findings are expected to promote research related to the molecular mechanism of ferroptosis in dairy cows, to provide support for the prevention and treatment of CM, and to contribute to the development of anti-inflammatory drugs based on HMOX1, FTH1 and iron metabolism.

The positive staining of ferric iron (brown) was apparent in MAC-T cells treated with LPS (100 μg/mL) and the ERA (5 μM) groups, while it was not present in the C (DMSO, 0.1% *v/v*) and NC groups (Figure 6A). TEM results suggested that the cristae and membranes of the mitochondria were intact and clear in the MAC-T cells of the DMSO and NC groups, while vanishing mitochondrial cristae and condensing of the mitochondrial membrane densities were observed in the MAC-T cells of the LPS and ERA groups (Figure 6B). ROS results displayed that the positive fluorescence signals were observed in four groups. The ROS signals in the ERA and LPS groups were obviously stronger than those in the DMSO and NC groups (Figure 6C). Compared to the DMSO and NC groups, the relative fluorescence intensity of ROS was up-regulated in the LPS and ERA groups with a significant difference (Figure 6D).

## 5. Conclusions

Ferroptosis was observed in the mammary glands of Holstein cows with CM and was positively correlated with the occurrence and development of CM. The ferroptosis process was triggered by up-regulation of HMOX1 and FTH in the mammary glands and iron accumulation. Similarity to ERA in the MECs, ferroptosis was induced by up-regulation of HMOX1 and FTH via iron-dependent metabolism, similar to the effect of ERA in MECs. Thus, the ferroptosis mechanism in the MECs of Holstein cows with CM was deduced in the present study. These results extend our understanding of ferroptosis and CM in Holstein cows, and will aid in the development of anti-inflammatory drugs to prevent and treat CM in diary animals.

## Figures and Tables

**Figure 1 antioxidants-11-02221-f001:**
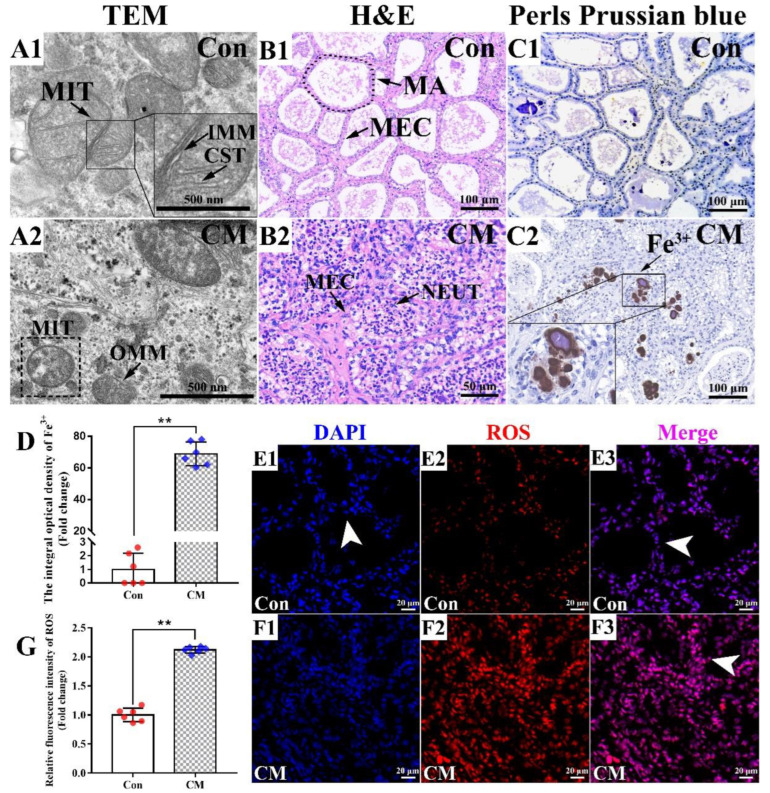
Phenotypes related to ferroptosis in the mammary glands of Holstein cows. (**A**) Morphology of the mitochondria was observed in the mammary glands of the control (Con) (**A1**) and clinical mastitis (CM) (**A2**) groups using TEM. (**B**) Morphological structures of the alveoli in mammary glands of the Con (**B1**) and CM (**B2**) groups were observed using H&E staining. (**C**) Observation of ferric iron (Fe^3+^) concentration variation in the mammary glands of the Con (**C1**) and CM (**C2**) groups using Perls plus DAB staining. (**D**) IOD values of Fe^3+^ in the Con and CM groups. (**E**,**F**), ROS immunofluorescent signals in the mammary glands of the Con (**E1**–**E3**) and CM (**F1**–**F3**) groups. (**G**) IOD values of the ROS in the mammary glands of the Con and CM groups. TEM, transmission electron microscopy. MIT, mitochondria. IMM, inner mitochondrial membrane. CST, mitochondrial cristae. OMM, outer mitochondrial membrane. H&E, hematoxylin–eosin staining. MA, mammary alveoli. MECs, mammary epithelial cells. NEUT, neutrophil. Con, control. CM, clinical mastitis. IOD, integral optical density. White arrow represents MECs. Scale bar of 500 nm, 50 μm, 100 μm and 20 μm represents 20,000×, 400×, 200× and 630×, respectively. Statistical analyses: Wilcoxon rank sum test, α = 0.05. Data are presented as median (rank). ** represents *p* < 0.01.

**Figure 2 antioxidants-11-02221-f002:**
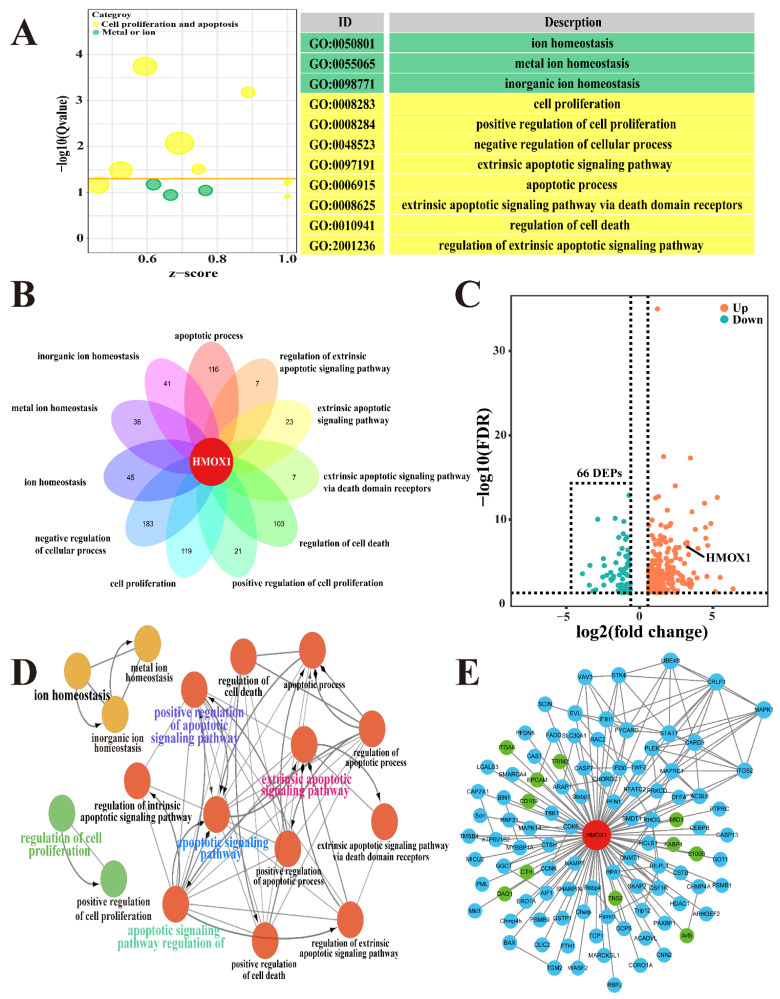
Identification of candidate DEPs associated with ion homeostasis, cell proliferation and apoptosis. (**A**) The significantly differential GO terms related to ion homeostasis, cell proliferation and apoptosis were selected from the DIA proteomics data. Green color represents the GO terms related to ion homeostasis. Yellow color represents the GO terms related to cell proliferation and apoptosis. (**B**) Venn diagram of the 11 GO terms selected from the DIA proteomics. (**C**) Volcano plot of the 302 DEPs included in the 11 GO terms. Blue color represents the down-regulated DEPs. Orange color represents the up-regulated DEPs. (**D**) The interaction network of the 11 GO terms related to ion homeostasis, cell proliferation and apoptosis. Green color represents the GO terms related to cell proliferation. Red color represents the GO terms related to cell apoptosis. Yellow color represents the GO terms related to ion homeostasis. (**E**) The PPI network of the 302 DEPs included in the 11 GO terms. Green color represents the 89 up-regulated DEPs. Blue color represents the 11 down-regulated DEPs. DEPs, differentially expressed proteins.

**Figure 3 antioxidants-11-02221-f003:**
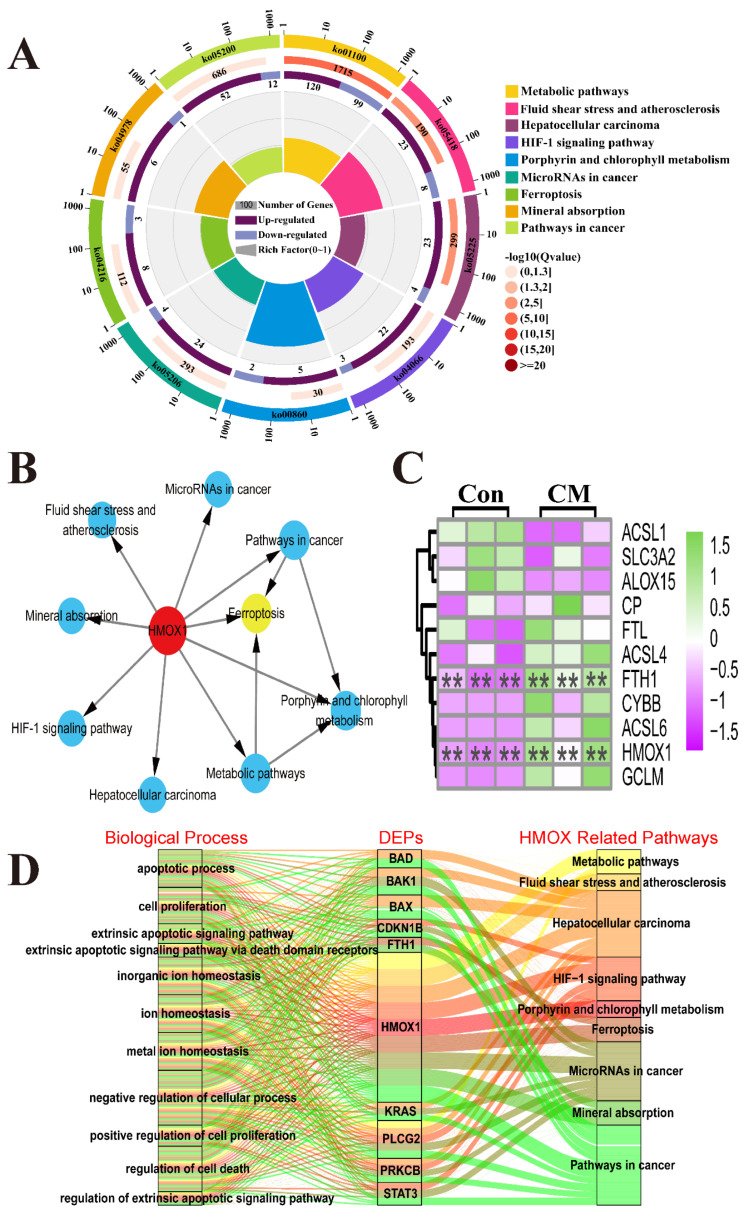
Identification of the pathways and candidate DEPs associated with HMOX1 based on the KEGG pathways of DIA proteomics. (**A**) Enrichment circle diagram of the significantly different pathways associated with HMOX1 based on the KEGG pathways of DIA proteomics. (**B**) The IPA network of the nine pathways associated with HMOX1. (**C**) The heat map of the ferroptosis pathway included 11 DEPs. (**D**) The Sankey diagram of the shared DEPs, the nine pathways and the 11 biological processes associated with ion homeostasis, cell proliferation and apoptosis. Con, control. CM, clinical mastitis. Statistical analyses: Wilcoxon rank sum test, α = 0.05. Data are presented as median (rank). ** represents *p* < 0.01.

**Figure 4 antioxidants-11-02221-f004:**
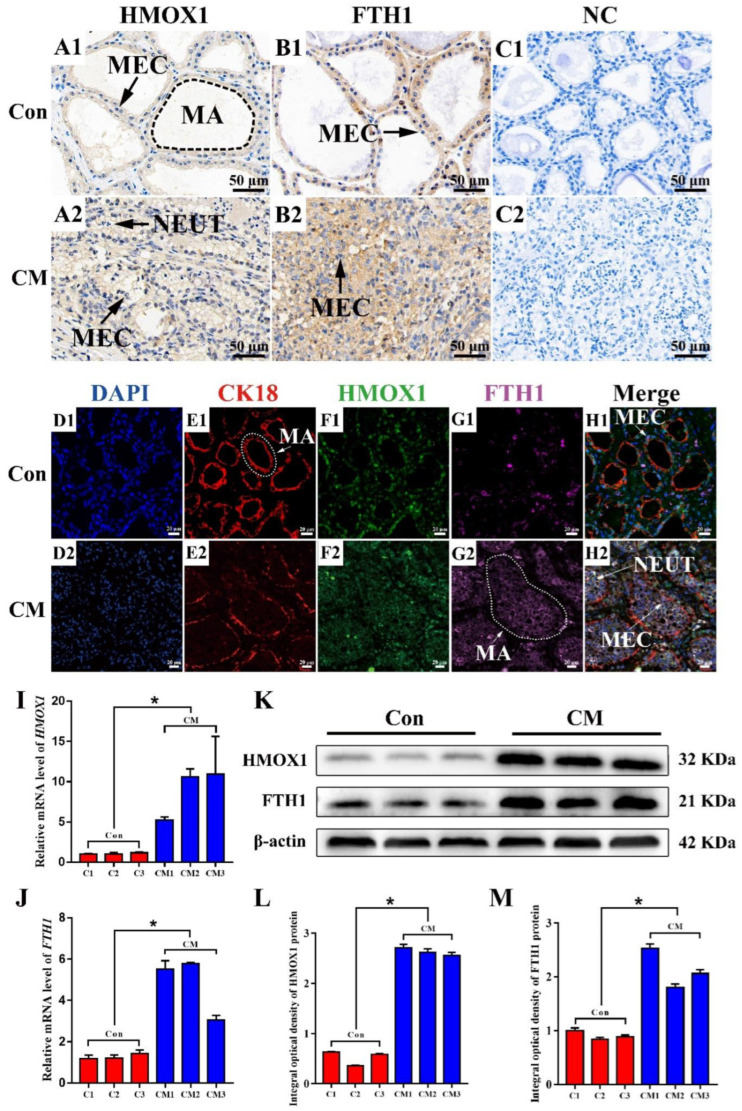
The distribution and expression pattern analysis of HMOX1 and FTH1 in the mammary glands of the Holstein cows. (**A**,**B**) The distribution analysis of HMOX1 (**A1**,**A2**) and FTH1 (**B1**,**B2**) in the mammary glands of the Con and CM groups using IHC staining. (**C**) Negative control HMOX1 and FTH1 in the mammary glands of the Con and CM groups. (**D**–**H**) Co-location analysis of DAPI (**D**), CK18 (**E**), HMOX1 (**F**), FTH1 (**G**) and the merged images in the mammary glands of the Con and CM groups using IF staining. (**I**,**J**) The expression pattern analysis of HMOX1 (**I**) and FTH1 (**J**) mRNA in mammary glands of the Con and CM groups using qRT-PCR. (**K**) Western blot expression pattern analysis of HMOX1 and FTH1 proteins in the mammary glands of the Con and CM groups. (**L**,**M**) The IOD values of HMOX1 (**L**) and FTH1 (**M**) proteins in the mammary glands of the Con and CM groups. MA, mammary alveoli. MECs, mammary epithelial cells. NEUT, neutrophil. IOD, integral optical density. Scale bar of 50 μm, and 20 μm represents 400× and 630×, respectively. Con/C, control. CM, clinical mastitis. NC, negative control. Statistical analyses: Wilcoxon rank sum test, α = 0.05. Data are presented as median (rank). * represents *p* < 0.05.

**Figure 5 antioxidants-11-02221-f005:**
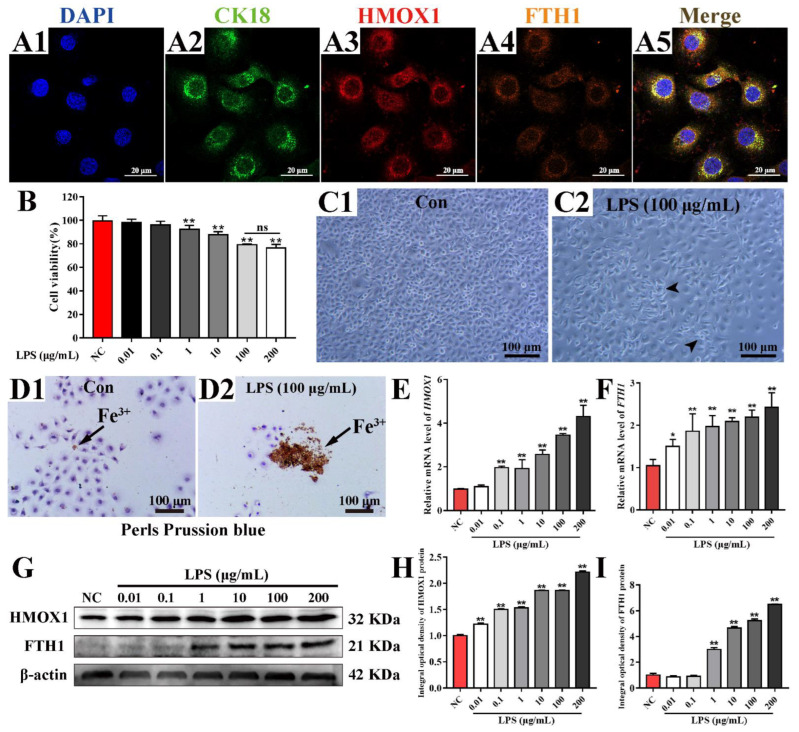
Up-regulation of HMOX1 and FTH1 with LPS induced inflammation in MAC-T cells. (**A**) Co-localization analysis of the HMOX1, FTH1 and CK18 proteins in the cytoplasm of MAC-T cells using IF staining. (**B**) Cell viability analysis of the MAC-T cells treated with LPS (0.01–200 μg/mL) using a CCK8 kit. (**C**) Morphological observation of the MAC-T cells treated with and without LPS (100 μg/mL). (**D**) Positive staining of ferric iron in MAC-T cells treated with and without LPS (100 μg/mL) using Perls Prussian blue staining. (**E**,**F**) The relative expression levels of HMOX1 and FTH1 mRNA in MAC-T cells treated with LPS (0.01–200 μg/mL) using qRT-PCR. (**G**) The Western blot expression analysis of HMOX1 and FTH1 proteins in MAC-T cells treated with LPS (0.01–200 μg/mL). (**H**,**I**) The IOD values of HMOX1 and FTH1 proteins in MAC-T cells treated with LPS (0.01–200 μg/mL). Con/C, control. LPS, experimental group, treated with LPS. NC, negative control. Statistical analyses: Student’s *t*-test. Data are presented as means  ±  SEM. * represents *p* < 0.05 and ** represents *p* < 0.01.

**Figure 6 antioxidants-11-02221-f006:**
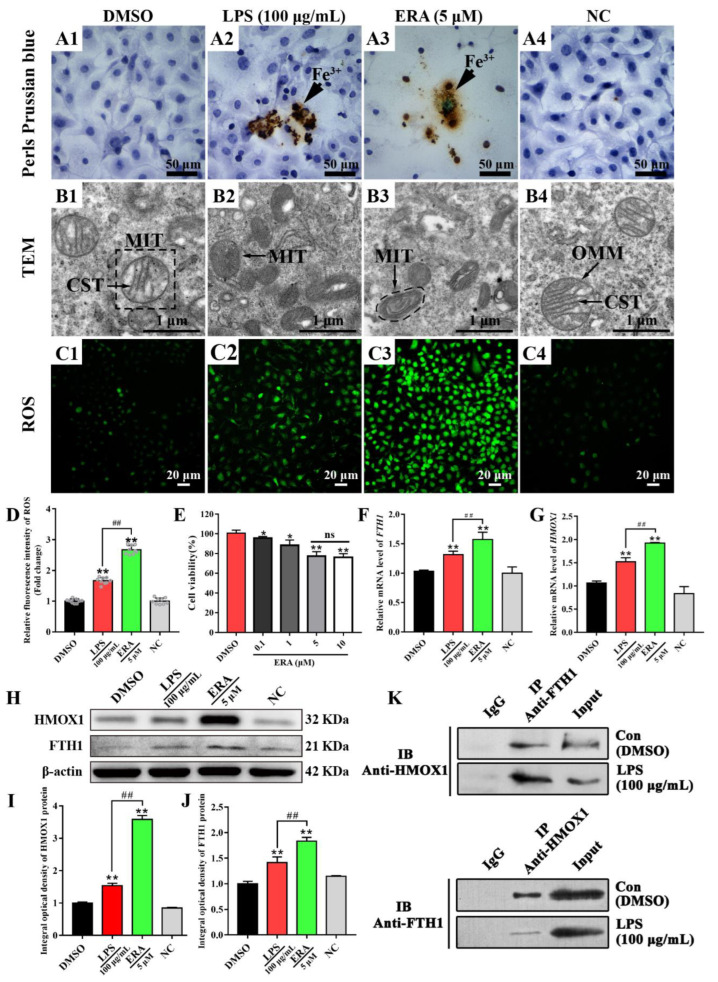
HMOX1 promotes ferroptosis via the FTH1 dependent pathway in LPS induced inflammation model in MAC-T cells. (**A**) Positive staining of ferric iron (Fe^3+^) in MAC-T cells treated with or without DMSO, NC, LPS (100 μg/mL) and ERA (5 μM) using Perls Prussian blue staining. (**B**) Morphology of the mitochondria was observed in the DMSO, NC, LPS and ERA groups using TEM. (**C**) ROS fluorescence signals in MAC-T of the DMSO, NC, LPS and ERA groups using DCFH-DA fluorescence probe. (**D**) The relative fluorescence intensity of ROS in the MAC-T of the DMSO, NC, LPS and ERA groups. (**E**) Cell viability analysis of MAC-T cells treated with ERA (0.1–10 μM) using a CCK8 kit. (**F**,**G**) Relative expression level analysis of HMOX1 and FTH1 mRNA in MAC-T cells treated with or without LPS (100 μg/mL) and ERA (5 μM) using qRT-PCR. (**H**) Western blot expression analysis of HMOX1 and FTH1 proteins in MAC-T cells treated with or without LPS (100 μg/mL) and ERA (5 μM). (**I**,**J**) The IOD values of HMOX1 and FTH1 proteins in MAC-T cells treated with and without LPS (100 μg/mL) and ERA (5 μM). (**K**) The Co-IP analysis of the FTH1 and HMOX1 proteins in MAC-T cells treated with and without LPS (100 μg/mL). Con/C, Control, treated with DMSO (0.1% *v/v*). LPS, experimental group, treated with LPS. ERA, experimental group, treated with ERA (5 μM). NC, negative control. IP, immunoprecipitation. IB, immunoblotting. Scale bar of 50 μm, 1 μm and 20 μm represents 400×, 15,000× and 400×, respectively. Statistical analyses: one-way ANOVA, α = 0.05. Data are presented as means  ±  SEM. * represents *p* < 0.05, ** or ## represents *p* < 0.01.

**Figure 7 antioxidants-11-02221-f007:**
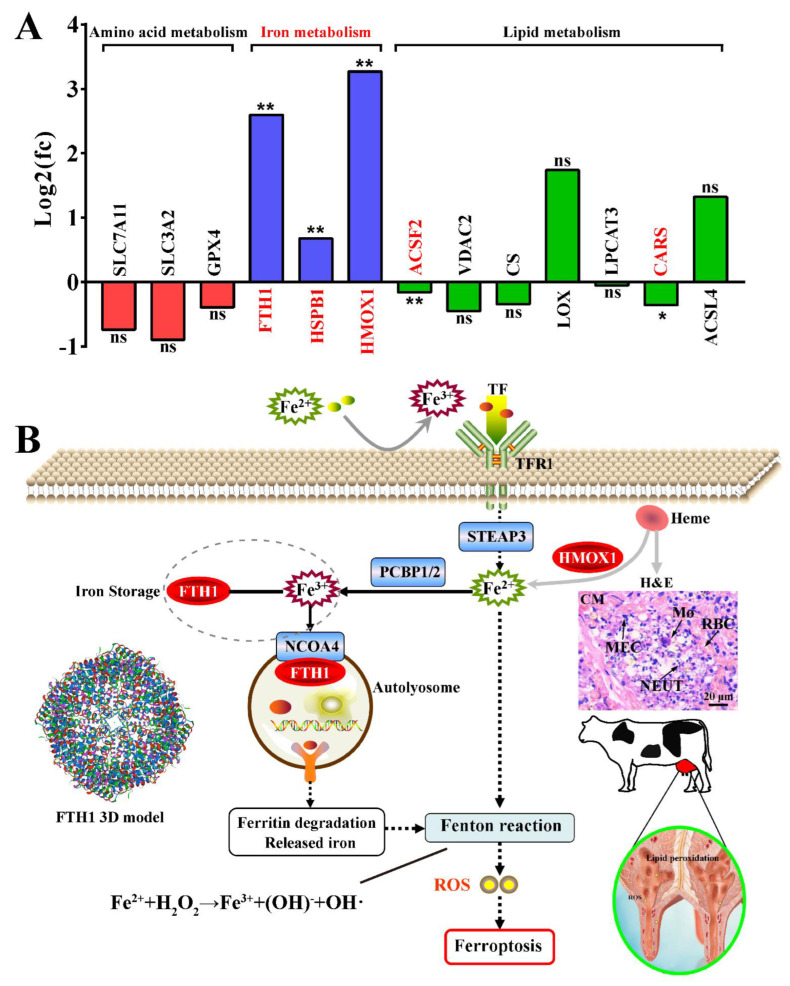
The deduced mechanism of HMOX1 promoting ferroptosis via FTH1 in the mammary glands of Holstein cows with CM. (**A**) The selected proteins associated with ferroptosis based on DIA proteomics. (**B**) The mechanism of ferroptosis in the mammary glands of Holstein cows with CM. MECs, Mammary epithelial cells. RBC, red blood cells. NEUT, neutrophil. M, macrophages. FC, fold change. CM, clinical mastitis. Scale bar of 20 μm represents 630×. Statistical analyses: Wilcoxon rank sum test, α = 0.05. Data are presented as median (rank). * represents *p* < 0.05, ** represents *p* < 0.01 and ns represents no significance.

## Data Availability

The data that support the findings of this study are available from the corresponding author upon reasonable request.

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
