# Peer review of "HMOX1 Promotes Ferroptosis in Mammary Epithelial Cells via FTH1 and Is Involved in the Development of Clinical Mastitis in Dairy Cows"

_antioxidants, 2022, doi:10.3390/antiox11112221_

Round 1
Reviewer 1 Report
This manuscript showed important results by analysis of HMOX1 in promotion of ferroptosis and involvement in the development of CM. These results can provide new insights for the prevention and treatment of ferroptosis-mediated CM in dairy animals. It can be basically acceptable to this Journal, however, the manuscript was written very poorly in English and data presentation as well as figure legends. These points should be extensively revised. Some examples were commented as below:
- Abstract:
English is very poor.
L16, its -> however, its
L20, GO -> full spell
L20-21, included -> including
L21-22, was identified and involved -> not incorrect, however, not proper in this style
L24-26, 'up-regulated' should not be duplicated. The sentences should be rewritten.
L29, You don't need to use abbreviation ERA, which is not appearing in abstract hereafter.
L30, clinical mastitis -> CM
In Figure formatting,
In Figure 1, A is not shown. Only A1 and A2 are shown. B, C, E, and F, too. You don't need to divide E and F. They can be shown with E.
L246, Morphology -> Morphologies
In Figure 2, Descrption -> Description
Figures 4, 5, and 6, Figure numbering should be simplified.
Reviewer 2 Report
This MS is very important to the field of clinical mastitis in dairy cows farming. It may shed new information of the importance of inflammation process that happen during lactation and may support treatments other than antibiotics to treat clinical mastitis. This is a preliminary study that I am sure it will be followed by many to provide important information on the CM in dairy cows.
Reviewer 3 Report
The manuscript submitted by Zhang et al. deals with the characterization of the role of ferroptosis in the development of clinical mastitis in dairy cows. The topic interesting and quite original, the language is clear and the methods applied are adequate. Tthe only criticism in my opinion concerns the number of animals involved in the experiment (n = 3 for both C and CM); generally the proteomics investigations refer to a minimum number of individuals equal to 4/5. The authors are called upon to justify their choice.
Specific comments:
- In the Abstract a brief description of the experimental design must be reported; it is not clear which results were obtained from the analyzes on animal tissue and which from cell culture
- L20: indicate the extended expression for GO.
- L196: have the extracted proteins been quantified? specify the method used and the amount of protein loaded for the electrophoretic run.
- L201: how the results were analyzed?
- In the paper the use of the Western blot is associated with the evaluation of the expression of the proteins of interest. In reality this is improper because this technique tells us how much protein is present in the sample but it is not certain that this data exactly correlates with the degree of expression at the gene level. This is the reason for which this assessment must be complemented with basic molecular biology assessments such as PCR. The authors are asked to revise some sentences of the text in accordance with this consideration.
Round 2
Reviewer 3 Report
All criticisms have been adequately addressed